# Establishment and Characterization of NCC-PMP1-C1: A Novel Patient-Derived Cell Line of Metastatic Pseudomyxoma Peritonei

**DOI:** 10.3390/jpm12020258

**Published:** 2022-02-10

**Authors:** Rei Noguchi, Yuki Yoshimatsu, Yooksil Sin, Takuya Ono, Ryuto Tsuchiya, Hiroshi Yoshida, Tohru Kiyono, Yutaka Yonemura, Tadashi Kondo

**Affiliations:** 1Division of Rare Cancer Research, National Cancer Center Research Institute, Tokyo 104-0045, Japan; renoguch@ncc.go.jp (R.N.); yyoshima@ncc.go.jp (Y.Y.); ishin@ncc.go.jp (Y.S.); takuono@ncc.go.jp (T.O.); rytsuchi@ncc.go.jp (R.T.); 2Department of Orthopedic Surgery, Graduate School of Medicine, Chiba University, Chiba 263-8522, Japan; 3Department of Diagnostic Pathology, National Cancer Center Hospital, Tokyo 104-0045, Japan; hiroyosh@ncc.go.jp; 4Exploratory Oncology Research and Clinical Trial Center, National Cancer Center, Kashiwa 277-8577, Japan; tkiyono@east.ncc.go.jp; 5NPO to Support Peritoneal Surface Malignancy Treatment, Japanese/Asian School of Peritoneal Surface Oncology, Kyoto 600-8189, Japan; y.yonemura@coda.ocn.ne.jp; 6Peritoneal Surface Malignancy Center, Department of Regional Cancer Therapy, Kishiwada Tokushukai Hospital, Kishiwada 596-8522, Japan; 7Peritoneal Surface Malignancy Center, Department of Regional Cancer Therapy, Kusatsu General Hospital, Shiga 525-8585, Japan

**Keywords:** pseudomyxoma peritonei, PMP, patient-derived cell line, SNP array, disseminated PMP mouse model, high-throughput drug screening

## Abstract

Pseudomyxoma peritonei (PMP) is the intraperitoneal accumulation of mucus due to a mucinous tumor. PMP predominantly occurs in low-grade carcinomas. The incidence rate of PMP is one to two cases per million people per year. The standard therapy of PMP comprises complete cytoreductive surgery and hyperthermic intraperitoneal chemotherapy. PMP recurs in about 50% of patients, and 30–40% are unable to receive the standard treatment because of its invasiveness. Therefore, novel therapies are of the utmost necessity. For basic and pre-clinical research, patient-derived cell lines are essential resources. However, only two PMP cell lines have been reported. Thus, we established a novel PMP cell line from resected metastatic PMP tissue. The cell line, named NCC-PMP1-C1, was maintained for more than 5 months and was passaged 25 times. NCC-PMP1-C1 cells demonstrated multiple amplifications and deletions, slow growth, tumorigenic ability, and dissemination of tumor cells in nude mice. We also used NCC-PMP1-C1 cells to screen drugs, which demonstrated a significant response to daunorubicin HCl, homoharringtonine, mitomycin C, and ponatinib. The NCC-PMP1-C1 cell line is the first PMP cell line derived from metastasized tissue and is a potential resource for basic and pre-clinical research on metastasized PMP.

## 1. Introduction

Pseudomyxoma peritonei (PMP) is the intraperitoneal accumulation of mucus derived from mucinous tumors [1]. PMP mainly originates from the appendix; however, it occasionally originates from other organs, including the ovaries [2,3,4,5], fallopian tubes [2], median urachus [3], colorectum [3,5], stomach [6], and pancreas [3,6,7]. PMP is a rare disease, with an incidence of one to two cases per million per year [8]. PMP shows various clinical manifestations due to the dissemination of mucin-producing tumor cells in the peritoneal cavity [9]. Histologically, PMP is divided into two groups: disseminated peritoneal adenomucinosis tumors (DPAMs), showing low-grade cytologic atypia and minimal architectural complexity, and peritoneal mucinous adenocarcinomas tumors (PMCAs), demonstrating high-grade cytologic atypia and/or complex epithelial proliferation. DPAM is reported to have association with a better prognosis than PMCA. The current standard therapy for PMP is complete cytoreductive surgery followed by hyperthermic intraperitoneal chemotherapy (HIPEC) [10]. However, distant metastasis is extremely rare in PMP [11,12], 50% of patients develop recurrence [13], 30–40% of patients are unable to receive the hybrid treatment because of its invasiveness [14], and patients with recurrent disease are difficult to cure [14]. Therefore, there is a pressing need to develop novel approaches to treat patients with PMP.

Patient-derived cancer models are essential resources to elucidate underlying molecular mechanisms of carcinogenesis and develop novel treatment strategies, as these models reproduce the genetic profiling and mimic the unique characteristics of the original tumor. Notably, patient-derived cell lines yield insightful information for researchers, facilitate screening, and help investigate a large number of anti-cancer drugs [15]. The large number of cell lines enables us to identify potential predictive biomarkers and detect therapeutic targets [16,17,18]. However, there is a significant dearth of cell lines of rare types of cancers, such as PMP, in public cell banks [19]. Therefore, the shortage of cell lines impedes basic and pre-clinical research. In particular, cell lines of specific cancers are required for cancer research for the following reasons: gene function is dependent on histological analysis [20], and the druggable mutations and the therapeutic targets of anticancer drugs are intricately associated [21]. Two cell lines (N14 and N15) of PMP have been reported [22] and are listed in the world’s largest cell line database, Cellosaurus [23]. However, they are not available in public cell banks. Thus, the establishment of additional patient-derived cancer cell lines for PMP is indispensable.

In our study, we established a novel cell line, NCC-PMP1-C1, derived from the resected metastatic tumor tissue from a patient with PMP. The utility of NCC-PMP1-C1 was evaluated in a drug screening of 214 anti-cancer drugs. We also examined the tumorigenic ability of NCC-PMP1-C1 in nude mice. To the best of our knowledge, this is the first study to establish a patient-derived PMP cell line originating from a metastasized site.

## 2. Materials and Methods 

### 2.1. Patient History

The patient was a 72-year-old man with a history of multiple pseudomyxoma peritonei (PMP) operations. Thirteen years prior to the cell line establishment, he was referred to Kusatsu General Hospital due to ascites. He underwent the first operation, composed of cytoreductive surgery (the completeness of the cytoreductive surgery: CCR was CCR-3 due to residual disease at the right diaphragm and the omentum minus), and intraperitoneal hyperthermic chemotherapy consisting of mitomycin C and cisplatin in 2007. Histological evaluation of the resected specimen showed high-grade pseudomyxoma peritonei. Seven years later, he underwent a second cytoreductive surgery, composed of resection of the residual tumors combined with peritoneal resection of the right diaphragm and pylorus gastrectomy. The patient presented to the hospital in 2020, and recurrence was suspected. The tumors eventually metastasized to the lower abdomen (Figure 1A) and right thigh (Figure 1B). The patient underwent resection of the metastatic thigh tumor. Histological evaluation of the resected specimen showed high-grade mucinous carcinoma peritonei (Figure 1C,D). After the surgery, chemotherapy consisting of irinotecan and oral S-1 was administered. A surgical resected tumor tissue of the metastatic thigh tumor was used to establish the cell line.

### 2.2. Cell Culture

Tumor cells were prepared as follows: resected tumor tissues were sectioned into small pieces (1–4 mm) and directly digested using the Tumor Dissociation Kit, Human MACS (Miltenyi Biotech, Bergisch Gladbach, Germany). The dissociated tissue was then passed through a sterile cell strainer (70 µm pore size). Enrichment of EpCAM-positive cells was performed using MACS microbeads (Miltenyi Biotech, Bergisch Gladbach, Germany) from the single-cell suspension. EpCAM-positive cells were seeded on Matrigel (0.16 mg/mL) and mouse embryonic fibroblast (MEF)-coated plates (BD Biosciences, San Jose, CA, USA). MEF isolated from E12.5 ICR12.5 mouse fetuses (Japan CLEA) were a kind gift from Dr. Akihiro Umezawa (National Institute for Child Health and Development, Tokyo, Japan). MEFs were cultured in Dulbecco’s modified Eagle’s medium (Nacalai Tesque, Inc., Kyoto, Japan) supplemented with 10% FBS and 1% penicillin—50-mg/mL streptomycin at 37 °C in a humidified atmosphere containing 5% CO_2_. When the cells at passage two reached subconfluence, the cells were incubated in the medium supplemented with 10 mg/mL mitomycin C for 4 h, and the cells were harvested and frozen with a TC protector (TCP-001, DS Pharma Biomedical, Osaka-shi, Japan) until use. Typically, the ELCA-positive cell density was maintained at 1 × 10^4^–2 × 10^4^ cells per cm^2^ in dishes of an appropriate size. Although the EpCAM-positive cells were seeded, mostly, after the first passage, fibroblast outgrowth was observed. Incubation with Accutase (Nacalai Tesque, Inc., Kyoto, Japan) for 1–2 min enabled the removal of fibroblasts; however, the dissociation of tumor cells was occasionally inefficient within the short duration, and the MACS EpCAM beads enabled us to resolve these cases. After more than two passages, when the fibroblasts were almost completely removed, cell seeding was performed at a density of 1 × 10^3^–2 × 10^3^ cells per cm^2^ based on the doubling time. The medium change was conducted at a regular interval of 3 days. After the establishment of the cell line, the cells were passaged at a ratio of 1:2 every 3–10 days. When a subconfluent state was reached, the cells were washed with phosphate-buffered saline (PBS; Nacalai Tesque, Inc., Kyoto, Japan). Then, the cells were detached with Accutase (Nacalai Tesque, Inc., Kyoto, Japan) and transferred to a fresh tissue culture plate. The cells were plated on Matrigel and MEF-coated culture dishes and maintained in Advanced DMEM/F12 (Life Technologies, Waltham, MA, USA) supplemented with a 2 mM L-alanyl-L-glutamine solution (Nacalai Tesque, Inc., Kyoto, Japan), Wnt3a/RSPO1 conditioned medium (prepared in-house), 10 mM nicotinamide (Wako, Kyoto, Japan), 10 µM p38 MAPK inhibitor SB-202190 (Selleck Chemicals, Houston, TX), 10 µM ROCK inhibitor Y-27632 (Selleck Chemicals, Houston, TX, USA), 10 µM TGF-β inhibitor DMH1 (Selleck Chemicals, Houston, TX), and 2% B27 (Thermo Fisher Scientific, Fair Lawn, NJ, USA). The medium was supplemented with GlutaMAX (Thermo Fisher Scientific, Fair Lawn, NJ, USA), 100 µg/mL penicillin, and 100 µg/mL streptomycin (Nacalai Tesque, Inc., Kyoto, Japan). To obtain Wnt3a/RSPO1-conditioned medium, LWnt3A cells (ATCC CRL-2647) were transduced with PQCXIP-RSPO1. The transduced cells were transfected with PQCXIP-NOG at an MOI of 10 for Wnt3a/RSPO1/Noggin conditioned medium.

### 2.3. Authentication and Quality Control

Short tandem repeat (STR) loci were examined using the GenePrint 10 system (Promega, Madison, WI, USA) for authentication of the established cell line, as previously described [24]. The STR patterns of the extracted DNA from both original tumor cells and established cells were investigated using GeneMapper software (Applied Biosystems, Foster City, CA, USA). The Cellosaurus database was used to compare the STR profiles of the cell line we established and cell lines previously deposited in public cell banks [23]. Mycoplasma contamination was detected using the e-Myco™ Mycoplasma PCR Detection Kit (Intron Biotechnology, Gyeonggi-do, Korea). PCR products were visualized on 1.5% agarose gels stained with Midori Green Advanced Stain (Nippon Genetics, Tokyo, Japan).

### 2.4. Genetic Analysis

*KRAS* mutations of the established cell line were examined following a previous report [25]. Briefly, according to the manufacturer’s instructions, reverse transcription of total RNA was performed using Superscript III reverse transcriptase (Invitrogen, Carlsbad, CA, USA). The *KRAS* was amplified with the forward primer KRAS_F (5′-TGT AAA ACG GCC AGT GTG TGA CAT GTT CTA ATA TAG TCA-3′) and the reverse primer KRAS_R (5′-CAG GAA ACA GCT ATG ACC GAA TGG TCC TGC ACC AGT AA-3′) using Platinum Taq DNA Polymerase High Fidelity (Thermo Fisher Scientific, Fair Lawn, NJ, USA). To analyze the Sanger sequencing data, an identical primer set for the junction and the BigDye Terminator v3.1 Cycle Sequencing Kit (Thermo Fisher Scientific, Fair Lawn, NJ, USA) were used in the Applied Biosystems 3130xL by GENEWIZ (GENEWIZ, South Plainfield, NJ, USA).

### 2.5. Single Nucleotide Polymorphism Array

Single nucleotide polymorphism (SNP) array genotyping was examined with Infinium OmniExpressExome-8 v. 1.4 BeadChip (Illumina, San Diego, CA, USA). Extraction of genomic DNA was conducted from the cultured cells derived from the tumor tissues, and the DNA was amplified. Amplified DNA was hybridized on array slides using an iScan system (Illumina, San Diego, CA, USA). The calculation of Log R ratios and B allele frequencies was performed using Genome Studio 2011.1, cnvPartition v3.2.0 (Illumina, San Diego, CA, USA), and KaryoStudio Data Analysis Software v. 1.0 (Illumina, San Diego, CA, USA). Annotation mapping was conducted using the human reference genome version hg19 (GRCh37). Analyses of the SNP array data were conducted using R software version 4.0.3 (R Foundation for Statistical Computing, http://www.R-project.org, accessed on 20 August 2023) and the R ‘DNAcopy’ package version 1.64.0 (Bioconductor, https://bioconductor.org/, accessed on 20 August 2023) [25] (the parameter: alfa = 0.0001). The whole-genome log 10 ratio (tumor/reference) value was smoothed, excluding chromosomes X and Y, and abnormal copy number regions were detected using the circular binary segmentation algorithm [26,27]. In the tumor cells, the definitions of amplifications and deletions were decided by regions with copy numbers >3 and <1, respectively. Genes that showed copy number alterations were annotated using the biomaRt package version 2.46.0 (Bioconductor, https://bioconductor.org/, accessed on 20 August 2023) and were queried for “cancer related genes” using the “Cancer Gene Census” in the Catalog Of Somatic Mutations In Cancer (COSMIC) database (GRCh 37 v91) [28].

### 2.6. Cell Proliferation Assay

Tumor cell proliferation was assessed using the Cell Counting Kit-8 (CCK-8; Dojin-do Molecular Technologies, Inc., Kumamoto, Japan). Assessments were done in triplicate. Briefly, the cells (5 × 10^3^) were seeded in 96-well plates coated with Matrigel and MEF. The cell number was assessed to measure the absorbance of samples in each well several times at 450 nm using a microplate reader (Bio-Rad, Hercules, CA, USA). By plotting the absorbance, we constructed growth curves. The growth curves were used to estimate the population doubling time.

### 2.7. Assessment of Tumorigenicity in Nude Mice

The animal experiments in this study were performed in compliance with the guidelines of the Institute for Laboratory Animal Research, National Cancer Center Research Institute. Briefly, female BALB/c nude mice purchased from CLEA Japan, Inc. (Tokyo, Japan) were utilized. A 100 μL volume of cells in a 1:1 mixture of Matrigel (BD Biosciences, San Jose, CA, USA) was intraperitoneally injected into the mice (1 × 10^6^ cells). Subsequently, observation of the tumor formation and measurement of the weight of the mice were performed weekly. After 2 months, the inoculated mice had developed abdominal distension, and the tumors were surgically resected for histological analysis.

### 2.8. Histological Evaluation

Tissues of the surgically resected tumor, pellets of the established cell line, and nodules from the mice were fixed in 10% neutral-buffered formalin for 24–72 h and embedded in paraffin. One representative 4 μm thick section of each specimen was stained with H&E, and another section was stained with the polyvalent basic dye Alcian blue (pH 2.5, 4085-2, Muto Pure Chemicals Co. Ltd., Tokyo, Japan). Other sections of the specimens were analyzed using immunohistochemistry (IHC). The following antibodies were used for IHC on the representative slides for each specimen: anti-cytokeratin 20 (CK20) (KS20-8, 1:50 dilution; Dako, Glostrup, Denmark), anti-CK7 (OV-TL12/30, 1:500 dilution; Dako, Glostrup, Denmark), anti-CDX2 (DAK-CDX2, prediluted; Dako, Glostrup, Denmark), and anti-SATB2 (A4B10, 1:1000 dilution; Abcam, Cambridge, MA, USA) antibodies. All IHC tests were performed using a Dako autostainer (Dako, CA, USA) according to the manufacturer’s recommendations. After deparaffinization, the tissue sections were stained using the antibodies described above and then counterstained with hematoxylin.

### 2.9. High-Throughput Drug Screening Test

A high-throughput drug screening test was performed following a previously reported method [24]. Briefly, NCC-PMP1-C1 cells were suspended in the PMP medium and seeded in a 384-well plate (Thermo Fisher Scientific, Fair Lawn, NJ, USA) coated with Matrigel and MEF at a density of 1 × 10^4^ cells using the Bravo automated liquid handling platform (Agilent Technologies, Santa Clara, CA, USA). On day 2, anti-cancer drugs (Selleck Chemicals, Houston, TX, USA; Appendix A) were administered at fixed or variable concentrations using the liquid handler. Cell proliferation was assessed 72 h later using the CCK-8 assay, according to the manufacturer’s instructions (Dojin-do Molecular Technologies, Inc., Kumamoto, Japan). Cell proliferation was evaluated from percent cell viability inhibition by comparison with the DMSO control cells. The IC_50_ value was calculated using GraphPad Prism 9.1.2 (GraphPad Inc., San Diego, CA, USA). All experiments were conducted in duplicate.

## 3. Results

### 3.1. Authentication and Quality Control of the Established Cell Line Derived from PMP Tissue

We established a cell line derived from a patient with metastatic pseudomyxoma peritonei (PMP). The cell line was named NCC-PMP1-C1. The cells grew semi-adherent for more than 5 months and passaged 25 times. Mycoplasma contamination was negative because mycoplasma-specific DNA was not detected in NCC-PMP1-C1 cells (data not shown). For authentication of the cell line, examination of 10 microsatellites (STRs) in the original tumor tissue and NCC-PMP1-C1 cells was conducted. The STR pattern of NCC-PMP1-C1 cells at all loci was similar to that of the original tumor (Table 1, Appendix A). The Cellosaurus database search showed that the STR pattern of NCC-PMP1-C1 cells did not correspond to that of other cell lines deposited in public cell banks. This finding confirms the NCC-PMP1-C1 cell line as a novel PMP cell line.

### 3.2. Characteristics of NCC-PMP1-C1 Cells

SNP array analysis revealed amplifications and multiple deletions in the original tumor tissue, and NCC PMP1-C1 cells were detected, which were similar (Figure 2). As copy number variants (CNVs), partial amplifications of chromosomes 1, 2, 7, 8, 9, 11, 12, 15, 16, and 17 and deletions of chromosomes 8, 10, 13, 14, 15, 17, and 20 (Appendix A) were involved. In NCC-PMP1-C1 cells, among the genes with multiple amplifications and deletions, CNVs of cancer-related genes were not identified. The KRAS c. G35T mutation, which is typical of PMP, was detected in NCC-PMP1-C1 cells (Figure 3). NCC PMP1-C1 cells exhibited a semi-adherent character and pleomorphic cell appearance (Figure 4A,B). The resuspended cells were embedded in paraffin for morphological observation. The consecutive sections were analyzed using H&E (Figure 4C,D) and Alcian blue staining (Figure 4E). The cells exhibited pleomorphic cells with nuclear atypia. The appearances matched the pathological features of the original tumor (Figure 1C,D). Alcian blue staining highlighted thick mucinous material with epithelial cells. The population doubling time of NCC-PMP1-C1 cells was approximately 147 h based on the growth curve (Figure 4F).

Sequencing data of *KRAS* showed the mutation peak in NCC-PMP1-C1 cells. The presence of a c.G35T mutation in the *KRAS* gene was confirmed in the cell line.

### 3.3. Tumorigenesis in Nude Mice

NCC-PMP1-C1 cells transplanted into BALB/c nude mice caused tumorigenesis under the described conditions. Mice carrying NCC-PMP1-C1 cells typically presented abdominal distension (Figure 5A), a gelatinous morphology with mucinous excrescence, and formation of tumor nodules on the serosal surface of visceral organs (Figure 5B, yellow arrowheads).

### 3.4. Histological Evaluation

The morphological features of the tumor tissue, the NCC-PMP1-C1 cells, and the tumor nodules of the nude mice were similar after H&E staining (Figure 6A,E,I). Cell atypia representing high-grade PMP histological features and extracellular mucin were observed (Figure 6A,E,I). Immunohistochemical staining revealed CK20, a marker of intestinal tumor and goblet cells. Caudal type homeobox2 (CDX2), a homeodomain transcription factor that is critical for MUC2 expression in intestinal goblet cells, and the special AT-rich sequence-binding protein 2 (SATB2), a marker of colorectal and appendiceal neoplasms, were examined. Immunohistochemical staining revealed similar expression and distribution patterns of CK20 (cytoplasmic, Figure 6B,F,J), CDX2 (nuclear, Figure 6C,G,K), and SATB2 (nuclear, Figure 6D,H,L). Taken together, these results suggest that NCC-PMP1-C1 and the tumor nodules of the nude mice recapitulated the morphological features of the tumor tissue.

### 3.5. Sensitivity to Anti-Cancer Drugs

The summarized evaluation of cell viability of NCC-PMP1-C1 after treatment with 214 anti-cancer drugs at 10 μM is depicted in Appendix A. Among the 214 anti-cancer drugs, 20 anti-cancer drugs that either showed significant anti-proliferative effects on NCC-PMP1-C1 cells or were frequently used as the standard therapy for PMP were further examined to calculate their IC_50_ values. The IC_50_ values of these anti-cancer drugs are listed in Table 2, and the growth curves based on the IC_50_ values are shown in Figure 7 and Appendix A.

## 4. Discussion

Pseudomyxoma peritonei (PMP) is an extremely rare disease involving the intraperitoneal accumulation of mucus due to mucinous neoplasia. PMP is associated with frequent recurrence and treatment complications. To develop novel therapy strategies, patient-derived PMP cell lines are essential. However, none are currently available. The rarity of PMP and the low-proliferating and high stroma-dependent features of PMP make the generation of PMP cell lines difficult. In this study, we reported a novel cell line, NCC-PMP1-C1, derived from the metastatic tumor tissue of a patient with PMP.

NCC-PMP1-C1 cells were established from a patient with high-grade PMP that metastasized to the right thigh. The strengths of this study are numerous. First, by collecting epithelial cells and optimizing the culture medium and matrix for PMP, we optimized protocols for the establishment of patient-derived PMP cell lines. Second, without artificial immortalization, such as transfection of the SV40 virus, we established NCC-PMP1-C1 with viability over 25 passages. Previously, two PMP cell lines, N14A and N15A, have been reported as the immortalized cell lines using a lentivirus with the entire SV40 genome, enabling the cells to achieve passaging for over 20 passages [29]. Third, NCC-PMP1-C1 cells comprised the first PMP cell line derived from a metastatic tumor of a patient with PMP. It is well known that PMP usually does not produce metastasis. NCC-PMP1-C1 originated from a distant metastasis and should be considered to be quite aggressive and not representative of the usual biological behavior of PMP. N14A and N15A are derived from primary appendiceal disease tissues [29]. Fourth, to validate NCC-PMP1-C1 cells for tumorigenesis, we injected NCC-PMP1-C1 cells into nude mice intraperitoneally. Cell-line-derived PMP mouse models were established to mimic tumor macro pathological findings, with abdominal distension, tumor nodule formation, mucin production, micro-pathological findings with cellular atypia, and similarity of immunohistochemistry. The cell-line-derived PMP mouse model can be used for the identification of anti-cancer drugs for HIPEC and provide suggestions for novel therapeutic strategies. Finally, the genetic profiles of PMP using SNP array analysis revealed multiple CNVs. Previously, Sio et al. reported MCL1 and JUN amplifications in several PMP samples using panel sequencing [30]. There are no reports of comprehensive copy number analysis with SNP array, CGH, whole exome sequence, or whole genome sequence in PMP.

We found that low concentrations of daunorubicin HCl, homoharringtnone, mitomycin C, and ponatinib had anti-proliferative effects on NCC-PMP1-C1 cells. Mitomycin C has been widely adopted for use in PMP treatment [31,32,33]. Previously, pharmacokinetic studies demonstrated a 107-fold increase in the concentration of mitomycin C in the intraperitoneal perfusate compared to plasma concentrations when administered systematically [31]. Sugarbaker et al. initially described the use of cytoreductive surgery and early postoperative intraperitoneal chemotherapy using mitomycin C [32,33]. Recent studies have reported the combination of cytoreductive surgery and intraoperative intraperitoneal chemotherapy under hyperthermic conditions [34,35,36]. Hyperthermia has been shown to potentiate the cytotoxicity of mitomycin C [37,38]. Our findings are consistent with these reports. In addition, the drug screening test also showed that even low concentrations of anti-cancer drugs such as daunorubicin HCl, homoharringtonine, and ponatinib effectively inhibited the proliferation of NCC-PMP1-C1 cells. Daunorubicin HCl, a topoisomerase II inhibitor that induces apoptosis, has been used in patients with hematological malignancies [39,40]. Homoharringtonine, a plant alkaloid extracted from *cephalotaxus*, has been approved by the US FDA for patients with hematological malignancies [41,42]. Ponatinib, a potent multi-target inhibitor of Abl, PDGFR, VEGFR2, FGFR1, and Src, has been used in patients with hematological malignancies [43]. However, these anti-cancer drugs have never been reported for PMP. Therefore, as the clinical utility, drug repositioning of these drugs for PMP is interesting, as the treatment of malignancies other than PMP using the identified anti-cancer drugs have been approved. The NCC-PMP1-C1 cell line may offer useful insights, elucidating the mechanism of efficacy of these anti-cancer drugs in PMP.

Several limitations of the current research should be addressed in future studies. First, the number of established cell lines should be expanded. We used a single cell line from only one patient with PMP. In oncology research, experiments based on a single cell line do not present conclusive results. Thus, validation in multiple cell lines originating from different samples is required. As PMPs are extremely rare, multi-institutional research will be necessary to additionally establish patient-derived PMP models. Second, the tumor tissues of PMP are composed of heterogeneous tumor cells as well as non-tumorous cells, including stromal cells and inflammatory cells. The established PMP cells may not demonstrate the inherent phenotypes and features due to the selection of the specific tumor cells through many passages. Furthermore, NCC-PMP1-C1 cells were potentially composed of heterogeneous tumor cells because cloning was not performed. Additional examination of NCC-PMP1-C1 cells will enhance their utility. Finally, we only examined the characteristics of NCC-PMP1-C1 cells under monolayer culture; thus, further examinations under various conditions, including spheroids and xenografts, would offer more insights into NCC-PMP1-C1 cells.

Patient-derived PMP cell lines are an important resource in oncology, and they ameliorate the investigation of gene function and evaluation of the efficacy of candidate drugs. Therefore, the NCC-PMP1-C1 cell line established in the current study may be a useful resource to analyze the molecular profiles of tumor progression and to develop new therapeutic strategies for PMP.

## Figures and Tables

**Figure 1 jpm-12-00258-f001:**
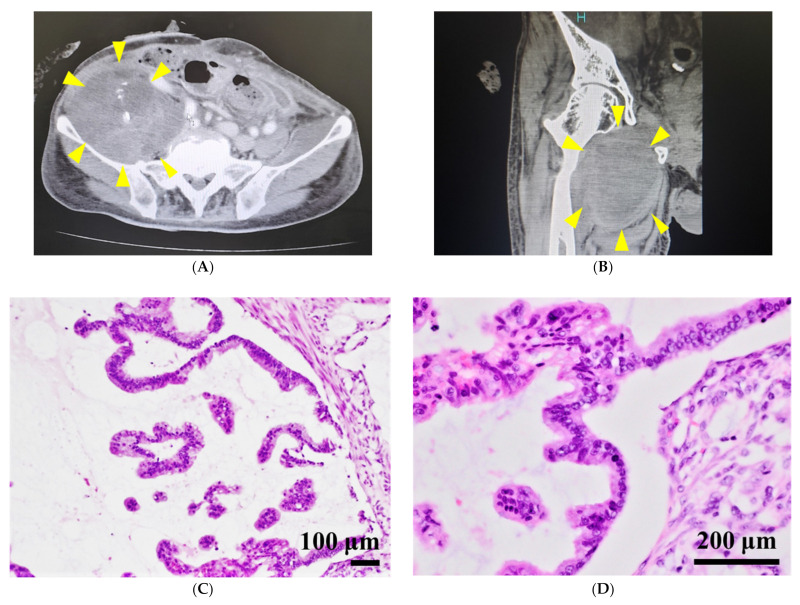
Clinical and pathological findings of the patient with metastatic PMP. (**A**) Intravenous contrast-enhanced CT scans showing a heterogeneously enhancing mass lesion (yellow arrowhead) in an area (diameter: 73 mm) in the right lower abdomen, and (**B**) a heterogeneously enhancing mass lesion in the patient’s right thigh. A hematoxylin and eosin-stained image of the original tumor. (**C**,**D**). Scale bars indicate 100 µm (**C**) and 200 µm (**D**).

**Figure 2 jpm-12-00258-f002:**
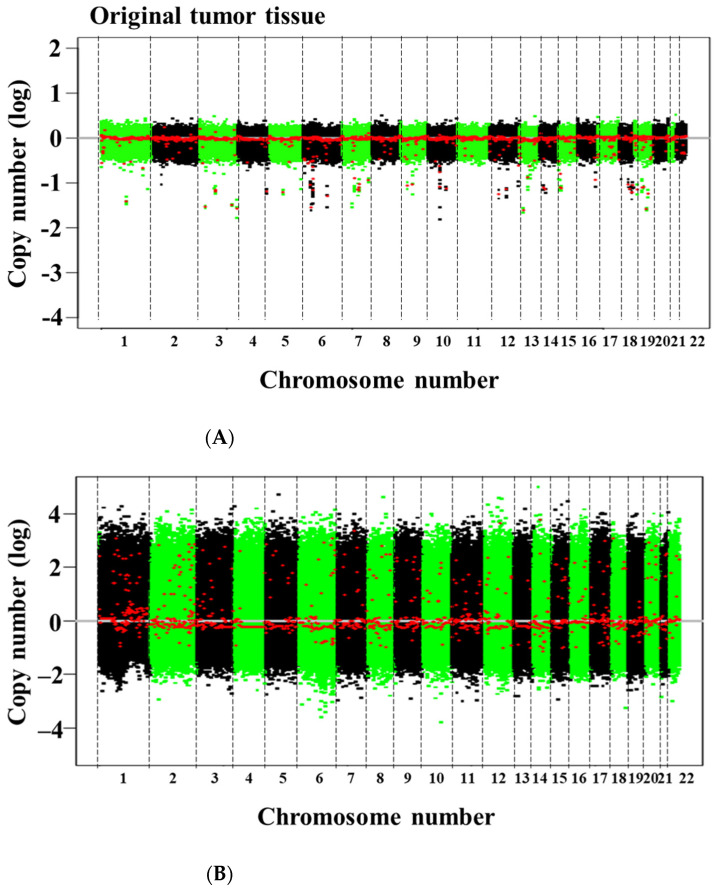
Single nucleotide polymorphism (SNP) array of the original tumor tissue and NCC-PMP1-C1 cells. The profile of copy number variants for the original tumor tissue (**A**) and NCC-PMP1-C1 cells (**B**). The x and y axes indicate the chromosome location and the log ratio of the copy number, respectively. The x and y axes indicate the genomic positions and the copy number, respectively. (**C**) The cytogenetic band of chromosome 19 and the copy number variation of chromosome 19. The x- and y-axes indicate the genomic positions and the copy number, respectively. The chromosomal segments are shown as red dots. The location of STK11 is shown with a red bar and arrow.

**Figure 3 jpm-12-00258-f003:**
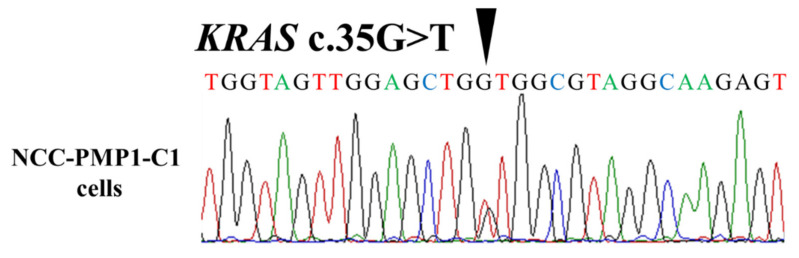
*KRAS* somatic mutation in NCC-PMP1-C1 cells.

**Figure 4 jpm-12-00258-f004:**
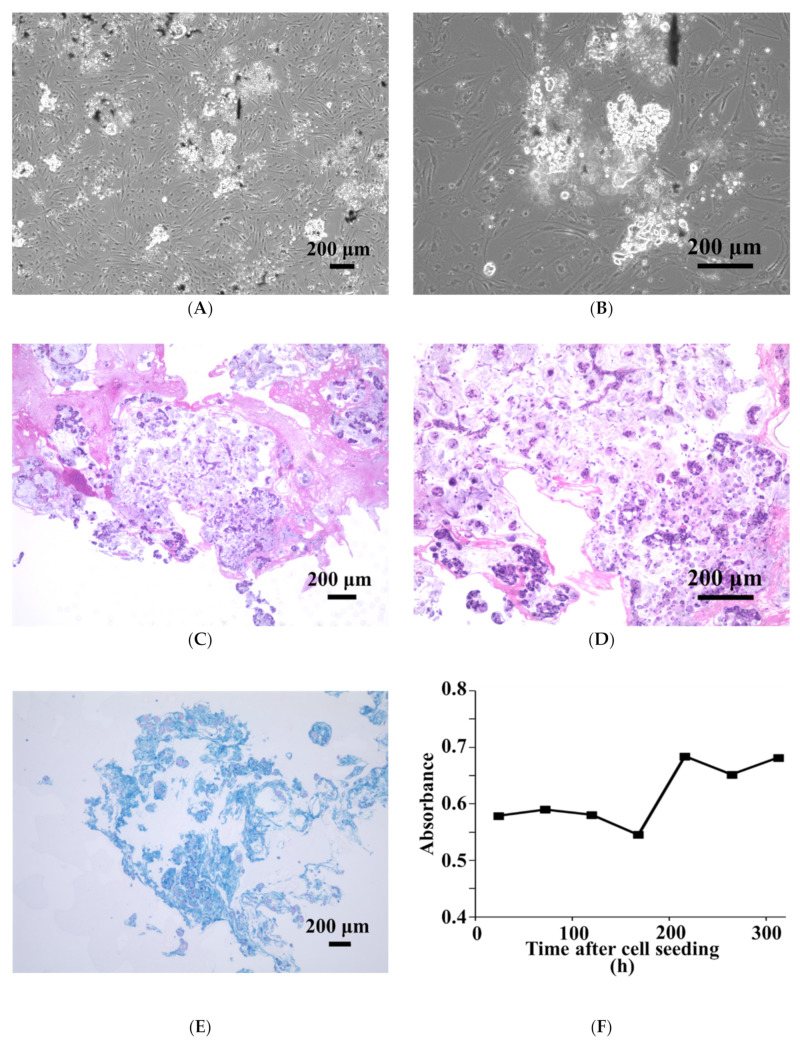
Characterization of NCC-PMP1-C1 cells. (**A**,**B**) NCC-PMP1-C1 cells exhibiting a pleomorphic morphology in phase-contrast micrographs. (**C**–**E**) NCC-PMP1-C1 cells were resuspended and embedded in agarose, then sectioned and examined using H&E (**C**,**D**) and Alcian blue staining (**E**). (**F**) Growth curve of NCC-PMP1-C1 cells. All scale bars indicate 200 µm.

**Figure 5 jpm-12-00258-f005:**
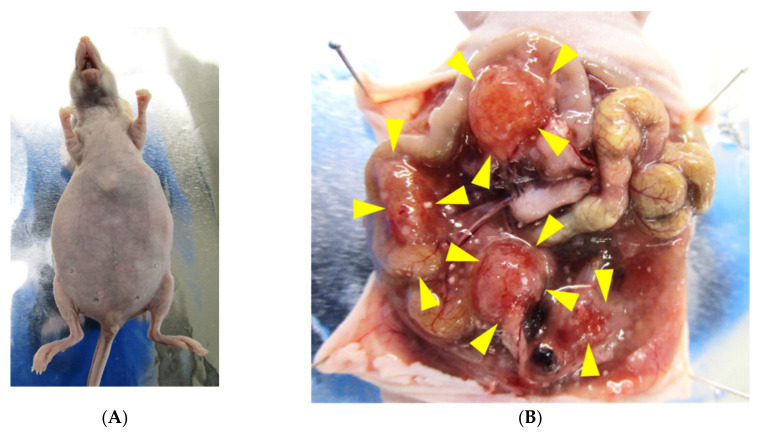
Tumorigenesis in nude mice. NCC-PMP1-C1 cells transplanted into BALB/c nude mice were capable of tumorigenesis. The mice typically presented abdominal distension (**A**) and a gelatinous morphology with mucinous excrescence (**B**). Yellow arrowheads indicate the presence of tumor nodules on the serosal surface of visceral organs (**B**).

**Figure 6 jpm-12-00258-f006:**
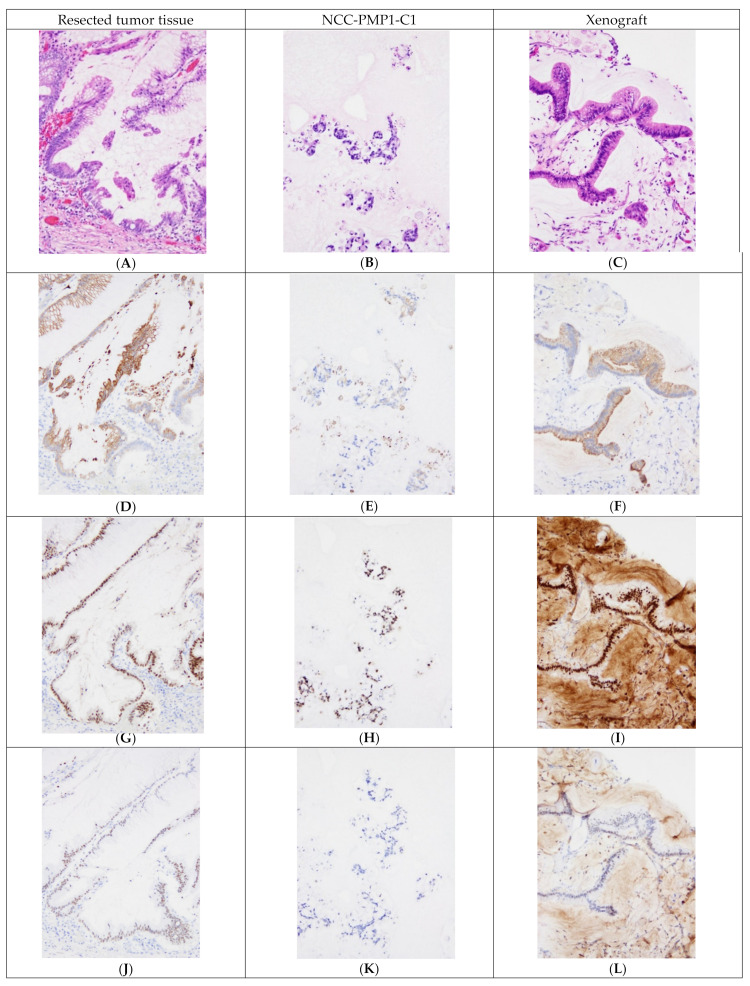
Representative histological and immunohistochemical findings of the surgically resected tumor (first column, ×200), corresponding established cell line (second column, ×200), and patient-derived xenograft (PDX) (third column, ×200). The first row (**A**–**C**) demonstrates the images of hematoxylin and eosin (H&E) staining; the second row (**D**–**F**) shows the images stained with anti-CK20 antibody; the third row (**G**–**I**) exhibits the images stained with anti-CDX2 antibody; and the fourth row (**J**–**L**) shows the images stained with anti-SATB2 antibody. In the surgically resected tumor (**A**), an appendiceal mucinous tumor composed of tumor cells showing dilated glands and papillary structure with abundant extracellular mucin is shown. These tumor cells were positive for CK20, CDX2, and SATB2 (**D**,**G**,**J**). The cell line established from the tumor (**B**) shows cohesive cell clusters with abundant extracellular mucous. Moreover, these cells were focally positive for CK20, CDX2, and SATB2 (**E**,**H**,**K**). The tumor from the PDX (**C**) was composed of mucinous tumor cells forming a papillary structure and small tumor cell clusters in the mucous pool. These tumor cells showed focal positivity for CK20 (**F**) and SATB2 (**L**) and diffuse positivity for CDX2 (**I**).

**Figure 7 jpm-12-00258-f007:**
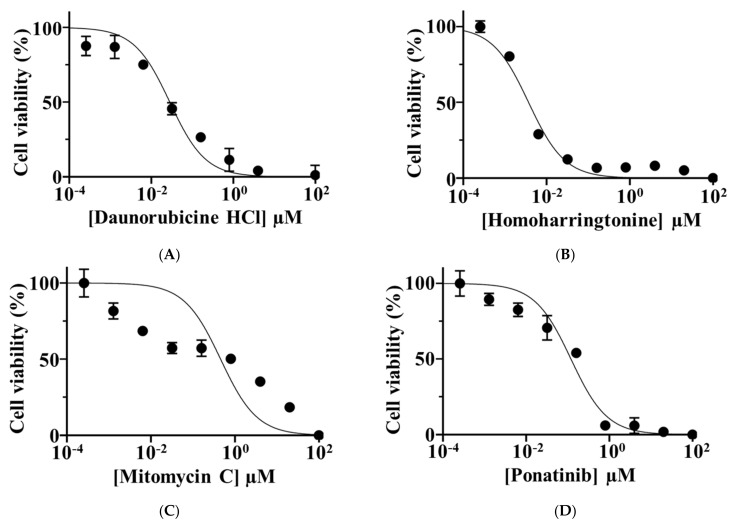
Viability of NCCPMP1-C1 cells treated with different concentrations of anti-cancer drugs. The cell viability at different concentrations of anti-cancer drugs was calculated based on growth curves for (**A**) daunorubicin HCl, (**B**) homoharringtonine, (**C**) mitomycin C, and (**D**) ponatinib. The x-and y-axes indicate the concentration of anti-cancer drugs and cell viability, respectively.

**Table 1 jpm-12-00258-t001:** Examination of short tandem repeat.

STR Locus (Chromosome)	Allele NCC-PMP1-C1	Allele Original Tumor Tissue
Amelogenin (X Y)	X, Y	X, Y
TH01 (3)	7	7
D21S11 (21)	30, 32.2	30, 32.2
D5S818 (5)	11	11
D13S317 (13)	9, 12	9, 12
D7S820 (7)	9, 11	9, 11
D16S539 (16)	10, 11	10, 11
CSF1PO (5)	11, 12	11, 12
vWA (12)	14, 17	14, 17
TPOX (2)	8, 11	8, 11

**Table 2 jpm-12-00258-t002:** List of half-maximal inhibitory concentration (IC_50_) values in NCC-PMP1-C1 cells.

CAS#	Drug	IC50 Value NCC-PMP1-C1 (µM)
179324-69-7	Bortezomib	1.732
1032900-25-6	Ceritinib	4.121
15663-27-1	Cisplatin	17.72
50-41-9	Clomifene citrate	7.746
220127-57-1	Crizotinib	11.85
23541-50-6	Daunorubicine HCl	0.0283
1108743-60-7	Entrectinib	2.119
21679-14-1	Fludarabine	31.63
51-21-8	Fluorouracil	15.76
129453-61-8	Fulvestrant	10.66
26833-87-4	Homoharringtonine	0.0037
136572-09-3	Irinotecan HCl	12.26
19767-45-4	Mensa	4.303
50-44-2	Mercaptopurine	22.88
50-07-7	Mitomycin C	0.456
61825-94-3	Oxaliplatin	8.761
943319-70-8	Ponatinib	0.1195
284461-73-0	Sorafenib	26.91
114899-77-3	Trabectedin	3.59
8918504-65-1	Vemurafenib	3.751

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
