# Peer review of "Establishment and Characterization of NCC-PMP1-C1: A Novel Patient-Derived Cell Line of Metastatic Pseudomyxoma Peritonei"

_jpm, 2022, doi:10.3390/jpm12020258_

Round 1
Reviewer 1 Report
This is a very interesting work reporting ex-vivo immortalization of a PMP cell line without using viruses. PMP cell lines are important to conduct preclinical experiments and, considering disease rarity and technical difficulties of having stable cell lines, Authors report deserves deep consideration. Study conceptualization is good with internal validation controls, such as genetic/SNP tests and murine model developement. Another strong point is the comparisong with IHC of original tumor, cell line and xenograft. Still there are some minor points in which Authors' explanations are needed in order to improve readability and comprehension.
2.1 Patient history
More informations would be helpful to determine precise clinical history.
-Probably using year of operation will make text easier to be read (e.g. first operation in 2007, second CRS in 2014).
-More details about operations are needed: I suppose cytoreduction was complete (CC0) in the first surgery having added HIPEC, whereas it was not in the second CRS (since HIPEC was not performed and term "disease progression" in the following lines instead of recurrence), correct? In case CRS was complete why HIPEC was not added (and in this case "progression" should be changed with more appropriate "recurrence")?
-did patient receive systemic chemotherapy before second surgery or after if cytoreduction was not complete? (I think he did not received after primary CRS, correct? But after iterative CRS, especially if not radical?).
- I would like to have more details about thigh localization that has been reported as distant metastasis. It is well known that PMP very rarely originates distant metastasis and tends to recur into the abdominal cavity. Could this localization be considered a diffusion through natural orifices (crural/inguinal hernia) or artificial (after two CRS that could have modified pelvic anatomy). Eventhough just seeing the reported CT-scan image, it actually seems a real distant metastasis (too deep into the thigh and close to femur to be an inguinal hernia). This point could be of interest for discussion (see later).
2.7 Assessment of tumorigenicity in nude mice
-it is not clear to the readers when and how tumor volume is calculated. It is assessed with radiologic techniques of the tumor mass every week during the two months? Or measures refer to mouse dimensions? Because the reported formula refers to an ellipsoid, so you considered the mouse whole abdomen?
3.4 Histological evaluation
-Figure 6 is not of immediate comprehension. I suggest to use a table-like configuration with headers to simplify it. For example row headers can be as follows first row: E&E staining, second row: anti-CK20, third row: antiCDX2... and the line headers first line: resected metastasis, second line: cell line, third: xenograft.
-Figure 6, image K. CDX2 is a nuclear marker, but comparing C and G images there is a lot of extracellular staining, is that "background noise"? is there a difference between the other two images that do not have such extracellulare staining?
- Discussion
-line 352. Lack of PMP lines is surely related to the disease rarity, but also to difficulties to obtain stable cell lines of a low-proliferating and highly-stroma-dependent tumor. I think this should be also considered (also taking in account the following points);
Considering additional informations required for clinical history (see above), I think Authors should add some reflections about potential selection biases after systemic chemotherapy (if any was administered), since systemic chemotherapy in patient history before excision of thigh metastasis could have selected clones resistant to used drugs, causing high IC50 values for such agents at in-vitro chemo-assays.
Another point to consider when proposing this cell line for preclinical tests, is that the sample comes from high grade PMP, that has different biological behaviour compared to low-grade PMP that are the majority of treated cases. If resected specimen originates from a real distant metastasis, this consideration becomes critical because PMP usually does not give metastases. So in the future experiments on this cell line, scientists should consider also the possibility of using a very aggressive tumor that does not represent the usual biological behaviour of PMP (this could explain previously unreported STK11 loss, or it could a novel interesting finding, further studies are needed).
Author Response
Comments and Suggestions for Authors
Reviewer1
Comment
This is a very interesting work reporting ex-vivo immortalization of a PMP cell line without using viruses. PMP cell lines are important to conduct preclinical experiments and, considering disease rarity and technical difficulties of having stable cell lines, Authors report deserves deep consideration. Study conceptualization is good with internal validation controls, such as genetic/SNP tests and murine model developement. Another strong point is the comparisong with IHC of original tumor, cell line and xenograft. Still there are some minor points in which Authors' explanations are needed in order to improve readability and comprehension.
Point 1: 2.1 Patient history
More informations would be helpful to determine precise clinical history.
-Probably using year of operation will make text easier to be read (e.g. first operation in 2007, second CRS in 2014).
Response 1: According to the reviewer’s comment, the year of operation was added to line 86-89 in the revised manuscript.
Point 2: -More details about operations are needed: I suppose cytoreduction was complete (CC0) in the first surgery having added HIPEC, whereas it was not in the second CRS (since HIPEC was not performed and term "disease progression" in the following lines instead of recurrence), correct? In case CRS was complete why HIPEC was not added (and in this case "progression" should be changed with more appropriate "recurrence")?
Response 2: I appreciated the reviewer’s comment. I detailed the operation from line 87 to 93 in the revised manuscript. I also added the explanation of the first CRS and changed from “progression” to “recurrence” in line 92.
Point 3: -did patient receive systemic chemotherapy before second surgery or after if cytoreduction was not complete? (I think he did not received after primary CRS, correct? But after iterative CRS, especially if not radical?).
Response 3: The patient did not receive systemic chemotherapy before second surgery or after primary CRS.
Point 4: - I would like to have more details about thigh localization that has been reported as distant metastasis. It is well known that PMP very rarely originates distant metastasis and tends to recur into the abdominal cavity. Could this localization be considered a diffusion through natural orifices (crural/inguinal hernia) or artificial (after two CRS that could have modified pelvic anatomy). Eventhough just seeing the reported CT-scan image, it actually seems a real distant metastasis (too deep into the thigh and close to femur to be an inguinal hernia). This point could be of interest for discussion (see later).
Response 4: I appreciate the reviewer’s interest in distant metastasis of our case. The typical blood-born distant metastasis in PMP is lung metastasis. However, metastasis found in his thigh may be the recurrence extending through inguinal ring from peritoneal metastasis, because he has peritoneal metastasis at the same time.
Point 5: 2.7 Assessment of tumorigenicity in nude mice
-it is not clear to the readers when and how tumor volume is calculated. It is assessed with radiologic techniques of the tumor mass every week during the two months? Or measures refer to mouse dimensions? Because the reported formula refers to an ellipsoid, so you considered the mouse whole abdomen?
Response 5: I appreciated the reviewer’s comment. We assessed the tumor mass formation every week during the two months and observed not tumor formation but abdominal distension. The reported formula was not used because we considered the mouse whole abdomen. Therefore, I corrected the methods in the manuscript (lines 207-212).
Point 6: 3.4 Histological evaluation
-Figure 6 is not of immediate comprehension. I suggest to use a table-like configuration with headers to simplify it. For example row headers can be as follows first row: E&E staining, second row: anti-CK20, third row: antiCDX2... and the line headers first line: resected metastasis, second line: cell line, third: xenograft.
Response 6: I appreciated the reviewer’s comment to make Figure 6 understandable for readers. I corrected the Figure 6 as a table-like configuration.
Point 7: -Figure 6, image K. CDX2 is a nuclear marker, but comparing C and G images there is a lot of extracellular staining, is that "background noise"? is there a difference between the other two images that do not have such extracellulare staining?
Response 7: Thank you for your insightful comment. As you pointed out, the extracellular staining appears to be non-specific. We anticipate the presence of a variety of necrotic cell-derived antigens in the highly viscous mucus, and there is no difference between C and G, as necrotic cell-containing mucus exhibits similar non-specific mucus staining in the parts not included in the figures.
Point 8: Discussion
-line 352. Lack of PMP lines is surely related to the disease rarity, but also to difficulties to obtain stable cell lines of a low-proliferating and highly-stroma-dependent tumor. I think this should be also considered (also taking in account the following points);
Considering additional informations required for clinical history (see above), I think Authors should add some reflections about potential selection biases after systemic chemotherapy (if any was administered), since systemic chemotherapy in patient history before excision of thigh metastasis could have selected clones resistant to used drugs, causing high IC50 values for such agents at in-vitro chemo-assays.
Response 8: Thank you for providing these insights. We agree with the reviewer’s suggestion that due to difficulties to obtain stable cell lines of a low-proliferating and highly-stroma-dependent tumor and modified the revised manuscript (lines 357-358). Against the second suggestion, in our case systemic chemotherapy was not administered. Therefore, the reflection about potential selection biases after systemic chemotherapy was not added to the revised manuscript.
Point 9: Another point to consider when proposing this cell line for preclinical tests, is that the sample comes from high grade PMP, that has different biological behaviour compared to low-grade PMP that are the majority of treated cases. If resected specimen originates from a real distant metastasis, this consideration becomes critical because PMP usually does not give metastases. So in the future experiments on this cell line, scientists should consider also the possibility of using a very aggressive tumor that does not represent the usual biological behaviour of PMP (this could explain previously unreported STK11 loss, or it could a novel interesting finding, further studies are needed).
Response 9: I appreciated the reviewer’s comment and added the sentence of the reviewer’s comment that NCC-PMP1-C1 originated from a distant metastasis and should be more aggressive that does not represent the usual biological behavior of PMP than low-grade PMP in line 369-371.
Reviewer 2 Report
I read with great interest this paper describing the establishment of a novel patient derived cell line NCC-PMP1-C1 .
I would like to understand if the cell line was developed at the first surgery metastasis or the second surgery at recurrence.
The histology mentioned was high grade, did it have any signet cell component?
How is this cell line different from colorectal metastasis? Will it be interesting to compare /find sililarities between colorectal and PMP of appendicular origin cell lines as these might have therapeutic implications especially if high grade and drug targets are studied
Author Response
I read with great interest this paper describing the establishment of a novel patient derived cell line NCC-PMP1-C1 .
Point 1: I would like to understand if the cell line was developed at the first surgery metastasis or the second surgery at recurrence.
Response 1: After the second surgery, the residual tumor was completely resected. Therefore, the cell line was developed at the second surgery metastasis.
Point 2: The histology mentioned was high grade, did it have any signet cell component?
Response 2: No, we did not detect any signet cell component in both surgical resected tumor and NCC-PMP1-C1 cells.
Point 3: How is this cell line different from colorectal metastasis? Will it be interesting to compare /find sililarities between colorectal and PMP of appendicular origin cell lines as these might have therapeutic implications especially if high grade and drug targets are studied
Response 3: I appreciated the reviewer’s comment. We will need the cell lines of colorectal originated PMP. To study therapeutic implications for colorectal metastasis, we will establish PMP cell lines originated from colorectal cancer in the future.
